# The Structure of *Clostridioides difficile* SecA2 ATPase Exposes Regions Responsible for Differential Target Recognition of the SecA1 and SecA2-Dependent Systems

**DOI:** 10.3390/ijms21176153

**Published:** 2020-08-26

**Authors:** Nataša Lindič, Jure Loboda, Aleksandra Usenik, Robert Vidmar, Dušan Turk

**Affiliations:** 1Department of Biochemistry, Molecular and Structural Biology, Jozef Stefan Institute, Jamova Cesta 39, 1000 Ljubljana, Slovenia; natasa.lindic@ijs.si (N.L.); jure.loboda@ijs.si (J.L.); aleksandra.usenik@ijs.si (A.U.); robert.vidmar@ijs.si (R.V.); 2Centre of Excellence for Integrated Approaches in Chemistry and Biology of Proteins (CIPKeBiP), Jamova Cesta 39, 1000 Ljubljana, Slovenia

**Keywords:** SecA2 structure, *Clostridioides difficile*, SecA2 secretory pathway, virulence factor export, ATPase, DEAD box, Walker motif, conformational change

## Abstract

SecA protein is a major component of the general bacterial secretory system. It is an ATPase that couples nucleotide hydrolysis to protein translocation. In some Gram-positive pathogens, a second paralogue, SecA2, exports a different set of substrates, usually virulence factors. To identify SecA2 features different from SecA(1)s, we determined the crystal structure of SecA2 from *Clostridioides difficile*, an important nosocomial pathogen, in apo and ATP-γ-S-bound form. The structure reveals a closed monomer lacking the C-terminal tail (CTT) with an otherwise similar multidomain organization to its SecA(1) homologues and conserved binding of ATP-γ-S. The average in vitro ATPase activity rate of *C. difficile* SecA2 was 2.6 ± 0.1 µmolPi/min/µmol. Template-based modeling combined with evolutionary conservation analysis supports a model where *C. difficile* SecA2 in open conformation binds the target protein, ensures its movement through the SecY channel, and enables dimerization through PPXD/HWD cross-interaction of monomers during the process. Both approaches exposed regions with differences between SecA(1) and SecA2 homologues, which are in agreement with the unique adaptation of SecA2 proteins for a specific type of substrate, a role that can be addressed in further studies.

## 1. Introduction

All bacteria possess systems for export of specific proteins from their site of synthesis in the cytoplasm to the cell envelope or extracellular environment. At the cell surface, these proteins play important roles in cell wall structure maintenance, nutrient acquisition and communication with the environment [1]. In bacterial pathogens, some of these surface proteins are also in an ideal position to interact with the host cells and are important virulence factors [2]. In general, most bacterial proteins are exported to the cell surface by the highly conserved general secretion (Sec) pathway through the cytoplasmic membrane-spanning SecYEG protein complex. For competent passage through the SecY channel, the C-terminal (mature) part of the protein destined for translocation must remain unfolded [3,4]. In most cases, the target protein is differentiated from the larger pool of cytoplasmic proteins by its specific N-terminal Sec signal peptide [5], which is recognized by the driving force of the Sec pathway, the SecA protein [6,7,8,9,10].

SecA is a cytosolic ATPase composed of multiple domains that undergo a series of conformational changes associated with the domain movement during the cycles of ATP binding and hydrolysis in the sequential translocation of the target protein through the SecY channel [11]. SecA is an essential housekeeping ATPase that has been well characterized in *Escherichia coli* and *Bacillus subtilis* and is important for the export of the majority of proteins to the periplasm [11]. Interestingly, in some commensal and food-grade bacteria as well as mycobacteria (*Mycobacterium tuberculosis* and *Mycobacterium smegmatis*) and a small subset of Gram-positive pathogens (*Listeria monocytogenes*, *Streptococcus gordonii*, *Staphylococcus aureus*, *Staphylococcus haemolyticus*, *Bacillus anthracis* and *C. difficile*) an accessory Sec system, characterized by SecA2, exists [2]. SecA2 proteins are similar to SecA proteins (called SecA1 in SecA2-possesing organisms), but are shorter in length and may contain deletions within one or more of their domains [2]. They are typically involved in the transport of a limited number of substrates with a major role in virulence or colonization [2]. In *C. difficile*, the leading cause of nosocomial diarrhea with common life-threatening complications worldwide, two Sec systems, characterized by SecA1 and SecA2, have been identified [12]. *C. difficile* SecA2 (CDSecA2) is involved in the export of the cell wall proteins (Cwps) that compose the outermost layer of *C. difficile*, the S-layer [12]. Among them is an important virulence factor, the SlpA protein. CDSecA2 knockdown leads to considerable growth defects and blocks translocation of Cwps, which accumulate in the cytoplasm of *C. difficile* cells [12]. CDSecA2 and *C. difficile* SecA1 (CDSecA1) share higher identity (54%) than most SecA2/SecA1 paralogue pairs of other species, i.e., the identity between SecA1/SecA2 pairs is 35%, 45%, 48%, 43% and 39% for *M. tuberculosis*, *L. monocytogenes*, *B. anthracis*, *S. gordonii* and *S. aureus*, respectively. Interestingly, SecA1 and SecA2 are both essential for *C. difficile* survival [12]. An essential function of the SecA2 system was only observed in one other bacterium, *Corynebacterium glutamicum* [13].

As often observed in SecA2-possessing species [14], CDSecA2 is encoded in the same locus as its major substrates, the Cwps [12]. The properties of Cwps seem to be sufficiently different from CDSecA1 substrates in that they cannot be translocated by SecA1 [12]. SecA2-specific set of substrates is also characteristic of *M. tuberculosis* and *L. monocytogenes* SecA2s [15,16,17], however, the mechanisms of these interactions are poorly understood.

The general understanding of roles of SecA1 and SecA2 in protein translocation has been established, however, insight in the specific differences called for structural analysis. We therefore determined two crystal structures of the CDSecA2 protein (apo and in complex with adenosine-5′-(γ-thio)-triphosphate (ATP-γ-S)). We confirmed the ATPase activity of CDSecA2 whereas the analysis combining sequential and structural data exposed importance of open conformation for dimerization and SecY binding, and sites relevant for further mechanistic studies of differences in interactions of SecA2 and SecA1 with their substrate and SecY channel.

## 2. Results

### 2.1. Except for the C-Terminal Tail Absence, the Structure of CDSecA2 Is More Similar to Its SecA(1) Homologues Than to M. tuberculosis SecA2

The full-length CDSecA2 protein with an additional SNAAA sequence at the N-terminus, which remained after proteolytic cleavage of the His tag, crystallized in the space group P2_1_2_1_2_1_. The crystals diffracted to a 2.3 Å resolution and contained one molecule per asymmetric unit (Table 1). The SecA2 chain was built from E13 to the C-terminal residue N781. Parts of the structure (residues E347, S348 and E627 and the region of K743-K748) without unambiguous interpretation of electron density maps were modeled with zero occupancy. The less ordered regions had higher B-factors (Appendix A). Table 1 contains the summary of data, refinement and model statistics for the structure. Structural data were deposited in the Protein Data Bank (PDB) with PDB ID code 6SXH.

The CDSecA2 crystal structure (Figure 1A) is a multiple domain structure similar to the previously studied SecA homologues (Figure 1B,C, Table 2). The ATPase part at the bottom of the CDSecA2 molecule is composed of two nucleotide-binding domains (NBD1 (E13 to T219 and Y356 to R395) and NBD2 (P396 to S577) shown in violet and green, respectively). They contain a central β-sheet with α-helices on both sides. Both domains contribute the residues that are important for catalysis. The primary structure of NBD1 is interrupted by the peptide cross linking domain (PPXD (P220 to T355), shown in red). The PPXD is attached to NBD1 with a pair of antiparallel β-strands forming a sheet with five hydrogen bonds. Above it, there is a central α-helix surrounded three 3 α-helices, a helix turn and a pair of antiparallel β-strands on each side with a highly conserved loop (V315-M323). The flexibility of the PPXD is evident from its higher B-factors in certain regions (residues G225-L231, N309, G310, and E343-N346; Appendix A).

The 66-Å-long α-helix of the helical scaffold domain (HSD (K578-N625), shown in orange) emerged from NBD2 and contacted the other four domains. HSD formed a three-helix bundle with the two parallel helices, known as the two-helix finger (2HF (G706-N781), shown in cyan) at the C-terminus of the molecule. The 2HF helices were connected by a 2HF loop of nine residues (L742-P750). The HSD also contained regions with higher B-factors (residues Q738-L742; Appendix A). Inserted into the three-helix bundle of the HSD is the helical wing domain (HWD (D626-L705), shown in yellow), composed of three α-helices and a loop (F667-S682). In our CDSecA2 structure, the PPXD was positioned near the HWD. It is a state known as the closed conformation [18,19,20,21,22].

In addition to closed conformation, the intermediate [23], open [7,19,26,27,28] and SecY-bound conformations [24,29,30] were observed, with PPDX occupying varying positions in regard to the HWD and different ligands. To present variations in positions of the PPXD, representative structures from different species were superimposed and are shown in Figure 1B. Their structural alignment RMSD and sequence-based comparison is shown in Table 2. Interestingly, CDSecA2 contained HWD (Figure 1C and Appendix A). This is similar to its paralogue, CDSecA1, and other SecA(1) proteins of the general SecA systems, but unlike many other SecA2 molecules. Namely, in the only SecA2 structure available to date, that of a pathogen *M. tuberculosis*, HWD was shorter (Figure 1C), as observed also in the other SecA2-encoding species (Appendix A). Importantly, in the structures of SecA-SecY(EG) complexes, HWD interacts with the SecY channel [24,29,30].

On the other hand, CDSecA2 lacked the C-terminal tail (CTT; Figure 1A). This was similar to the other SecA2 proteins and opposed to SecA(1) proteins (Appendix A). In *E. coli* CTT truncations of SecA lead to translocation defects [31,32]. CTT in general SecA systems interacts with the ribosome [33], co-translationally recognizes nascent substrate proteins [33,34] as well as binds in the substrate binding groove and affects conformation of the PPXD [35]. While proteins targeted for the general SecA must be maintained in unfolded conformation [11], SecA2 targets may be folded previously to export [36], implying an absence of SecA2-ribosome interaction and altogether supporting the differential SecA1 and SecA2 export mechanisms.

Altogether, the absence of CTT in CDSecA2 suggests a different mechanism of target protein recognition and SecA activity modulation compared to SecA(1)s, in contrast to probable SecY interaction and target translocation similarities mediated through conserved regions of shared domains.

### 2.2. CDSecA2 Is an Active ATPase In Vitro

To confirm that the clostridial SecA2 protein indeed hydrolyses ATP, we used an ATPase assay based on the malachite green reagent. The reagent reacts with the free phosphate released after the enzymatic reaction, resulting in the colorimetric product that is proportional to the enzymatic activity. The average rate of ATP hydrolysis of CDSecA2 was 2.6 ± 0.1 µmolPi/min/µmol CDSecA2, which is similar to the ATP hydrolysis rate of *M. tuberculosis* SecA2 [37].

### 2.3. The Structure of CDSecA2 in Complex with ATP-γ-S Shows Conserved Regions for Nucleotide Binding

The crystal of the CDSecA2 and ATP-γ-S complex diffracted considerably worse than the apo structure with a notable anisotropic component. Correspondingly, its structure was less well resolved. Nevertheless, the electron density map for the ATP-γ-S ligand unambiguously positioned it in the active site (Figure 2A,C). Table 1 contains the summary of data, refinement, and model statistics for the structure. Structural data were deposited in the Protein Data Bank with PDB ID code 6T4H.

The structures of apo CDSecA2 and CDSecA2-ATP-γ-S complex were very similar (Figure 1 and Figure 2A) as they aligned with RMSD 0.85 Å (Table 2). Binding of ATP-γ-S did not lead to any considerable conformational change such as opening of the PPXD. Crystal packing of CDSecA2 requires contacts between PPXD of one and NBD1 and NBD2 of the adjacent SecA2 molecule, therefore larger conformational changes of PPXD would disrupt the crystal lattice. As evident from the CDSecA2-ATP-γ-S complex structure (Figure 2A,C), the highly conserved residues involved in ATP binding and hydrolysis were positioned within Walker motifs A and B at the interface of NBD1 and NBD2, and are present in all SecA homologues, including CDSecA1 and CDSecA2 (Figure 2B and Appendix A). The first aspartate (D207) in the Walker B motif of CDSecA2 is in close proximity to the gamma phosphate group of ATP-γ-S, which superimposes with the catalytic magnesium coordinated by two water molecules in the *B. subtilis* SecA complex with ADP (PDB ID 1M74 [18]). As in another SecA [27] the opposite side of the gamma phosphate moiety of ATP-γ-S contains the conserved lysine (K106) in Walker motif A in both SecA1 and SecA2 in *C. difficile*, and the mutation of this lysine is lethal, suggesting its essential role of both systems [12]. In *E. coli*, *M. tuberculosis* or *M. smegmatis* [37,39,40], mutation of the equivalent lysine abolishes ATP binding and the biological activity of ATPase. Mutations of residues in Walker motif B in *E. coli* and *B. subtilis* allow the SecA protein to bind, but not hydrolyze, ATP [41,42,43]. From the CDSecA2 structure with the bound nucleotide and residue conservation in structural alignments, we thought that the requirements for ATP binding and hydrolysis were likely similar to those of other SecA ATPases.

### 2.4. Movement of CDSecA2 Domains Affects the Conformation of the Target Protein Binding Site

To provide insight into the feasibility of transitions between closed, open and SecY-bound conformations of the PPXD, the transitions were modeled (Figure 3A). For the open conformation representative of the SecA in contact with the target protein [19], we used *B. subtilis* SecA with a bound peptide as a template (PDB ID 3JV2) [19]. Superimposition indicated an 82° movement and 20 Å displacement of the PPXD away from the HWD and towards NBD2 (Figure 3A). This movement created the clamp that captures the protein through the rotation of the PPXD, analogous to the structure of SecA in a peptide-bound state [19]. Our model suggested that movement of the PPXD was similar for CDSecA2. For the SecY-bound conformation, we used the structure of *T. maritima* SecA bound to the SecYEG channel as a template (PDB ID 3DIN) [24]. The two helix turns in the first helix of the PPXD were unwound due to a lack of three residue insertion in the CDSecA2 sequence in the P220-L236 region compared to the longer *T. maritima* SecA sequence of the P265-Q284 region (Figure 3A and Appendix A). This model showed a 122° movement and 25 Å displacement of the PPXD towards NBD2, leaving the highly conserved loop (V315-M323; cyan in Figure 3A and Appendix A) in close contact with the nucleotide binding site. A 17° movement and 1.8 Å displacement of the HWD and the HSD-2HF region was also evident (yellow in Figure 3A). In addition, in the *T. maritima* SecA structure, the helix I400-Y409 (corresponding to Q357-E364 in the CDSecA2 structure) that links NBD1 with the PPXD was two residues longer and had a different fold and position, which prevented us from analogously connecting our CDSecA2 PPXD to the remainder of the molecule and modeling the β-strands connecting the PPXD to NBD1 (Figure 3B). Importantly, this region (Figure 3B,C and Appendix A) includes residues previously shown to be involved in the interaction of general SecAs with the signal peptides [6,7,8,9,10] and nascent proteins [19,44,45], thus indicating a possible CDSecA2 specific structural feature.

## 3. Discussion

Crystal structures show that CDSecA2 has a fold similar to its homologues, with multidomain organization consisting of NBD1, NBD2, PPXD, HSD and HWD, but lacks CTT. Analysis supports a hypothesis that during ATP hydrolysis and substrate binding CDSecA2 undergoes a transition from the closed to the open conformation that enables transportation of the substrate through the SecY channel. Separate binding sites for the protein substrate and SecY channel and a catalytic site for ATP are required for this process. Using our CDSecA2 structure, we attempted to reveal similarities and differences between the SecA1 and SecA2-dependent protein transport systems.

First, we inspected the ATP binding site. The basic requirements for ATPase activity seem to be highly conserved among various bacterial species (Figure 2 and Appendix A). These include not only the ATP binding motifs (Figure 2 and Appendix A), but also the PPXD loop (depicted in cyan in Figure 3A and Appendix A) that contacts the nucleotide binding site during PPXD conformational changes and triggers ATP hydrolysis [46,47]. This indicates that the mechanism that couples the protein binding and passing through the SecY channel with the ATPase binding and hydrolysis cycles is conserved in all SecA proteins, including CDSecA2.

Next, we inspected the structural differences between CDSecA2 and its homologues. The CDSecA2 is in possession of the whole length HWD, which is in contrast to many other SecA2-encoding species, like *Mycobacterium* and *Corynebacterium* species (Appendix A), where HWD is shorter. Other Gram-positive SecA2 proteins also appear to have smaller deletions/insertions in their HWDs when compared to the general SecA systems (Appendix A). In the HWDs of the general SecA systems, residues important for target protein binding were identified [6,9,48]. Furthermore, HWD interacts with C-terminal part of the SecY in crystal structures of SecA-SecY(EG) complexes [24,29] and with adjacent lipids [30]. Thus, the lack of HWD in SecA2 systems may enable binding of specific substrates by increasing solvent exposure of the target recognition site [23] and hamper binding of SecA2 to SecYEG. We showed that the conformation of CDSecA2 HWD closely resembles those of the general SecA systems (Figure 1C), thereby suggesting CDSecA2 may bind to SecYEG and participate in substrate recognition and translocation in a similar manner. Altogether, it seems that even though SecA2 systems are designated for export of specific substrates, the mechanism/s they acquire for this action may vary.

Since a number of studies have shown that dimeric SecA at least in part mediates protein translocation in general secretion systems [49,50,51,52,53] we also focused on requirements for the CDSecA2 dimerization. With respect to the PPXD position and the extent of interactions between one SecA2 molecule and crystallographic symmetry-related molecules, apo and ATP-γ-S-bound CDSecA2 crystallized as a closed monomer. So far, PPXD has been found in various conformations in crystal structures (Figure 1B and references within). This is similar to HWD, that can rotate by up to 15° in different crystal structures [26] (Figure 3A), but may assume alternative conformations in solution [6,30,54,55]. Moreover, the modeled CDSecA2 dimers (Figure 4A,B) revealed that neither closed SecA(1/2) PDB deposits nor SecY-bound *B. subtilis* SecAs modified with either a substrate peptide insertion in the 2HF tip [29] or C-terminal nanobody extension [30] are able to form such dimers, because only the open and unmodified SecY-bound conformations, that are essential for substrate peptide binding [19] and protein translocation [24,29,30], enable SecA dimerization through HWD/PPXD cross-interactions. This suggests that this dimeric conformation is physiologically relevant.

Next, we turned our attention towards specificity for substrate and SecY interactions. To expose the differences that might facilitate specific interactions, we performed evolutionary conservation analysis of their surfaces. We first prepared a model of CDSecA1 in the closed and open conformation using the SWISS-MODEL server [56], using the structures of *Thermus thermophilus* SecA (PDB ID 2IPC) [57] and *B. subtilis* SecA (PDB ID 3JV2) [19] as templates, respectively. Then, both open models, CDSecA2 and CDSecA1, and the SecA2 and SecA1 sequences from thirty different SecA2-encoding organisms were analyzed using the ConSurf server [58].

The survey showed that the binding sites for the target proteins are conserved among SecA2 and SecA1 proteins; however, similarity at the interacting surface regions between the target proteins and the SecA2 proteins in the SecA2-encoding organisms was smaller than that of the SecA1 proteins (Figure 5). This applies to the signal peptide binding site and also to the nascent protein binding area (the clamp). The target protein binding regions were positioned in the groove between NBD1 and NBD2 and the PPXD, which contains residues important for signal peptide recognition (Figure 3C, Figure 5 and Appendix A) [6,7,8,9,10]. They are followed by the residues involved in nascent protein interaction in the clamp (depicted with an arrow in Figure 3C and Figure 5) [19,44,45]. Our analysis was consistent with the findings of our modeling attempt, which among the available structures of SecY-bound SecA conformation lacked a suitable template for the interaction site of the target protein that connects the two β-strands of the PPXD to NBD1 in CDSecA2 (Figure 3A,B and Appendix A).

In addition, the SecY channel interaction region is located at the 2HF loop and its surrounding area [23,24,59], where high surface residue conservation suggests their conserved functional role (Figure 3C and Figure 5 and Appendix A). Namely, superimposition of our CDSecA2-SecY-bound conformation model with the *T. maritima* SecA-SecY structure (PDB ID 3DIN; [24]) showed structural similarities that would lead to the PPXD with its loops and the long helix of the HSD forming the contact surface with the SecY channel and the 9-residue-long and flexible 2HF loop inserting into the SecY channel (residues L742-P750; Figure 3C and Appendix A). The key regions of CDSecA2 that were modeled to interact with the *T. maritima* SecY pore were also conserved (Appendix A). The identified residues in CDSecA2 (T319 in the PPXD, L402, V406 and S437 in NBD2, and R613 in the HSD) corresponded to those from *E. coli* SecA that were found to interact with SecY [59]. Our analysis, however, exposed differences among the residues proposed to be involved in signal transduction of the correctly bound signal peptide to the SecY channel, thereby activating the translocation process [8]. Among them is residue S226, which in *E. coli* SecA lies in the β-strand that connects the PPXD with NBD1 and has also been implicated in SecY interaction [59] and nascent protein binding in *B. subtilis* SecA [19]. This serine residue at the corresponding position is conserved in all SecA1 proteins, including CDSecA1 (S224; Figure 5 and Appendix A). However, in CDSecA2 and the majority of other SecA2 homologues, the serine has been replaced by alanine (A224). Since the mutation of this serine to alanine in *E. coli* SecA reduced the secretion of the target protein [8], the serine/alanine difference indicates a difference in the mechanism of signaling of the presence of the bound peptide between the SecA1 and SecA2 proteins and consequently indicates differences in signal transduction and interaction with the SecY channel.

This interpretation is further supported by the identified differences among the sequences of the 2HF loop, where a major nascent protein contact site is provided by Y745 in CDSecA2. This residue can be replaced by another hydrophobic residue (V799 in CDSecA1; Figure 5), as the side chain sits in the hydrophobic pocket in SecY [45]. There is also a 2HF helix-terminating proline present in CDSecA2 (P750), as in the majority of SecA homologues (Appendix A). However, this helix-terminating proline is missing in organisms with SecA2 systems with a designated SecY2 channel (streptococci, staphylococci, bacilli and lactobacilli). In some cases, these organisms also lack the starting leucine (L742 in CDSecA2) and, importantly, the interacting tyrosine or the bulky hydrophobic residue at the homologous position (Appendix A).

To summarize, the analyses presented here identified regions that contribute to the mechanism of SecA2-specific target recognition and interaction with the SecY channel. Our modeling exposed importance of the open conformation for formation of dimer, which can bind the SecY. An immediate suggestion from this study is that the locking PPXD domain in closed or transient conformation by a compound binding in the hinge region connecting PPXD domain to NBD1 may provide a new target site for drug discovery. Moreover, we were unable to find a template to model this region in the open conformation (Figure 3). This suggests that a ligand binding in that area may specifically inhibit the translocation of proteins in *C. difficile* SecA2 pathway, and due to the almost identical sequence in this regions, also its SecA1 homologue. Not so surprisingly, this further suggests that mechanistic differences between SecA1 and SecA2 should be studied pairwise. Nevertheless, our analysis (Figure 5) exposed lower sequence conservation in the CDSecA2 substrate loading region (encircled with blue line), and differences in 2HF finger, which pinpoint to further mutation studies to be tested in a substrate and SecY binding assay. Hence, we expect that the knowledge gathered in this study will enable us to address the question of how the SecA1 protein functions a part of the “housekeeping” export system in bacteria and, furthermore, how the SecA2 protein functions as a part of the translocation system for specific, possibly folded protein substrates [2,3,4,6,7,8,9,10,36].

## 4. Materials and Methods

### 4.1. CDSecA2 Protein Expression and Purification

The gene encoding CDSecA2 (UniProt: Q183M9; KEGG ID: CD630 27920) was cloned into the pMCSG7 ligation-independent vector [60] and overexpressed in *E. coli* BL21 (DE3) cells. Cultures were grown in 2 L of ZYM-5052 autoinduction medium with 50 mg/mL ampicillin as described by [61]. Cells were harvested and lysed, after which the lysate was separated by centrifugation and purified by HisTrap FF and HiPrep 26/60 Sephacryl S-200HR (GE Healthcare Life Sciences, Chicago, IL, USA) columns. Fractions of the resulting recombinant protein, CDSecA2 (781 residues, 89.1 kDa) containing the native sequence with the N-terminal hexa-histidine tag, and the additional sequence for protease cleavage were combined, and a recombinant His6-tobacco etch virus protease (1:50 (*w/w*)) was used to remove the His tag over the course of a 2-day incubation at 4 °C with 10 mM β-mercaptoethanol. After a secondary IMAC step using Ni Sepharose 6 Fast Flow (GE Healthcare Life Sciences, Chicago, IL, USA), the flow-through fraction containing the protein of interest was collected and concentrated in 30 mM Tris, 400 mM NaCl (pH 7.5), 10 mM imidazole and 10 mM β-mercaptoethanol to 2.3 mg/mL. For cocrystallization with the ATP-γ-S, β-mercaptoethanol was omitted.

### 4.2. Crystallization and Structure Determination of CDSecA2

CDSecA2 crystals were obtained using the vapor-diffusion technique in sitting drops by mixing a 1 µL aliquot of the protein solution with 1 µL of reservoir solution containing 0.1 M BisTRIS propane, pH 8.3, 0.2 M sodium acetate and 18% *w/v* PEG3350, followed by drop equilibration at 20 °C over 350 µL of reservoir solution. Cocrystallization of ATP-γ-S and CDSecA2 was set up in the same manner with a reservoir solution that also contained 4 mM MgCl_2_ and 2 mM ATP-γ-S. In both cases, rod-shaped crystals typically appeared within 1-2 days. Crystals were harvested after a week and flash cooled in liquid nitrogen. For identification of CDSecA2, the crystals were dissolved in ultrapure water and digested with trypsin. The samples were analyzed by LC–MS/MS using an EASY-nanoLC II HPLC unit (Thermo Fischer Scientific, Waltham, MA, USA) coupled to an Orbitrap LTQ Velos mass spectrometer (Thermo Fischer Scientific, Waltham, MA, USA). Protein identification was performed using the MaxQuant software package version 1.6.3.4 integrated with the Andromeda search engine [62]. Native data sets were collected at a wavelength of 0.9184 Å by BMX14.1 (BESSY, Berlin, Germany) and processed with the XDS Software [63]. Datasets of CDSecA2 cocrystallized with ATP-γ-S were collected at a wavelength of 1.0000 Å by XRD2 (Elettra, Trieste, Italy) and processed with XDS software [63]. SecA2 from *B. subtilis* (PDB ID 1TF2; [26]) was successfully used in phasing by molecular replacement with Phaser software [64]. MAIN software [65] was used for the subsequent steps of structure determination for map calculation, model building, refinement, validation and deposition. During refinement, the maximum likelihood free kick target function was applied. The idea behind the Rkick is similar to the Rfree, however, it is calculated from the structure factors of the kicked model against WORK set of data and not from structure factors of unperturbed model against TEST part of data. This corrects the conceptual problem of Rfree. Namely, the claim that TEST set used in cross validation maximum likelihood refinement target function as a source of structure independent information is false because errors in models under refinement are not randomly distributed [38]. Geometric restraints of ATP-γ-S were generated by PURY [66]. Three-dimensional images were generated with MAIN using RASTER 3D rendering software [67] and PyMOL software (The PyMOL Molecular Graphics System, Version 1.8, Schrödinger, NY, USA).

### 4.3. ATPase Activity Assay

A malachite green assay (Sigma-Aldrich, St. Louis, MO, USA) was used to measure the ATPase activity of CDSecA2. Three independent experiments of triplicate wells of a 96-well plate were set up containing either 2, 4 or 6 µL of 0.75 mg/mL CDSecA2 in 30 mM Tris and 400 mM NaCl pH 7.5, and the samples were brought to a final volume of 10 µL with assay buffer (40 mM Tris, 80 mM NaCl, 8 mM Mg(CH_3_COO)_2_ and 1 mM EDTA, pH 7.5). Reaction mixes composed of 20 µL of assay buffer and 10 µL of 4 mM ATP were added to each sample, background blank and negative control well. The reactions were incubated for 30 min. Formation of inorganic phosphate was monitored spectrophotometrically by the increase in absorbance at 620 nm at 30 min after adding 200 µL of malachite green reagent. The inorganic phosphate concentrations generated in the reaction mixtures were calculated using a standard curve. The assay was conducted at 25 °C, and the average rate of ATP hydrolysis and the standard error were determined.

### 4.4. Prediction of Evolutionary Conservation of SecA2 Proteins

The ConSurf server [58] was used to estimate the evolutionary conservation of amino acid residue positions within the SecA2 and SecA1 proteins (Figure 5). Analyses were performed using the coordinate files for the open conformation CDSecA2 and CDSecA1 models. The CLUSTAL format alignment file for 30 SecA2 and 30 SecA1 sequences and the phylogenetic tree were constructed by the ConSurf server with a neighbor-joining algorithm. Conservation scores were obtained using the WAG model of substitution and the Bayesian method of calculation. Structure-based SecA sequence alignment was performed with the PROMALS3D server [68] (Appendix A) using the same sequences as in Consurf analyses, and additional *T. maritima*, *E. coli* and *B. subtilis* SecA sequences for orientation.

### 4.5. Modelling of Dimer in Open and YEG-Bound Conformation

Models of open and YEG-bound CDSecA2 were superimposed with FATCAT [25] to both molecules of the *B. subtilis* SecA dimer (PDB ID 3JV2) and *T. maritima* SecA1 (PDB ID 3DIN), respectively.

## Figures and Tables

**Figure 1 ijms-21-06153-f001:**
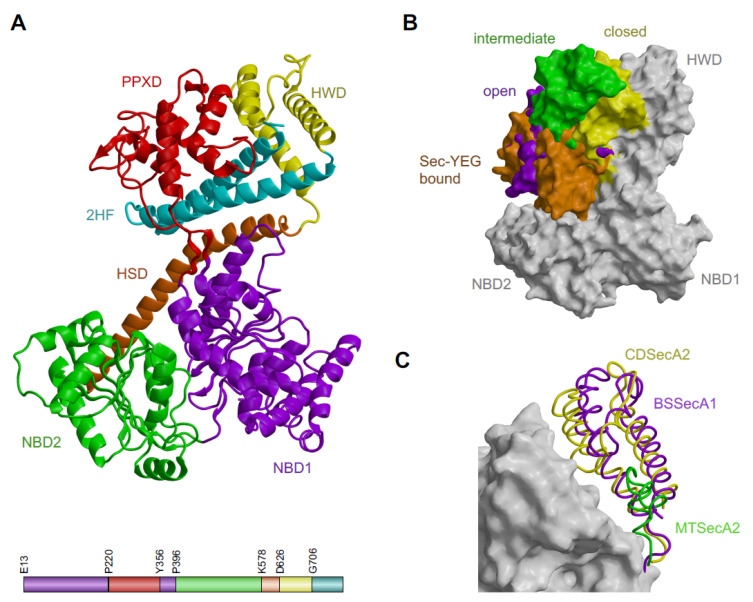
The crystal structure of apo CDSecA2 and conformational variability of SecA homologues. (**A**) The CDSecA2 crystal structure in closed conformation. Domains are color coded: violet, nucleotide-binding domain 1 (NBD1); green, nucleotide-binding domain 2 (NBD2); red, peptide cross linking domain (PPXD); orange, helical scaffold domain (HSD); cyan, two-helix finger (2HF) and yellow, helical wing domain (HWD). Locations of the domains in the linear sequence (bottom) define the color coding. The residues at the beginning of the domains are depicted. (**B**,**C**) The structures of different SecA molecules were superimposed onto CDSecA2 (PDB ID 6SXH) shown as white surface. (**B**) The colored surface of PPXDs in different conformations. Closed, intermediate and open conformations of CDSecA2 (PDB ID 6SXH), *M. tuberculosis* SecA2 (PDB ID 4UAQ) [23] and of *B. subtilis* SecA (PDB ID 3JV2) [19], are shown in yellow, green and violet, respectively. Sec-Y(EG) bound conformation of *Thermotoga maritima* SecA (PDB ID 3DIN) [24] is shown in orange. (**C**) The HWDs of different sizes presented as ribbons. Yellow, HWD of CDSecA2 (PDB ID 6SXH); violet, HWD of *B. subtilis* SecA1 (PDB ID 1M6N) [18] and green, HWD of *M. tuberculosis* SecA2 (PDB ID 4UAQ) [23].

**Figure 2 ijms-21-06153-f002:**
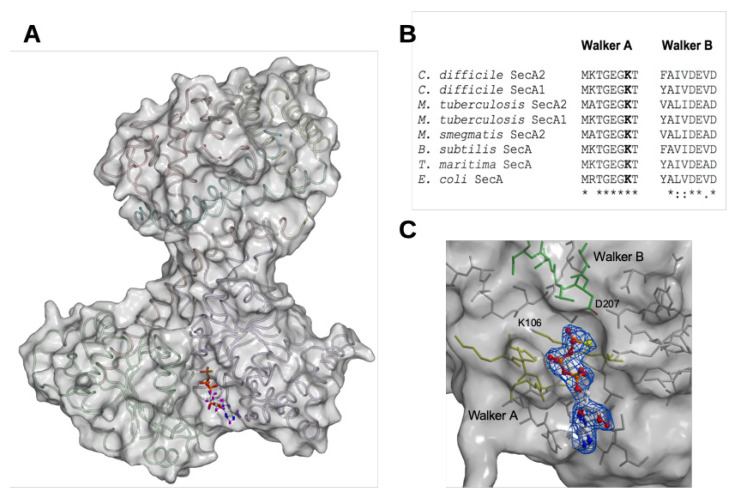
CDSecA2 ATP binding. (**A**) The structure of CDSecA2 (transparent surface with ribbons) in complex with ATP-γ-S (sticks). (**B**) The conservation of Walker motifs in the CDSecA1 and A2 proteins is shown in sequence alignment. Conserved residues of CDSecA2 involved in nucleotide and Mg binding and ATP hydrolysis are aligned with the corresponding residues of CDSecA1 and other SecA representatives from various bacterial species. (**C**) The CDSecA2 surface region that binds ATP-γ-S (stick and ball representation) in maximum-likelihood free kick weighted 2mFo-Dfc [38] is depicted. Walker A and Walker B motifs are shown as yellow and green sticks, respectively. Color code of ATP-γ-S: blue, nitrogen; red, oxygen; white, carbon; orange, phosphate; yellow, sulfur and purple, hydrogen.

**Figure 3 ijms-21-06153-f003:**
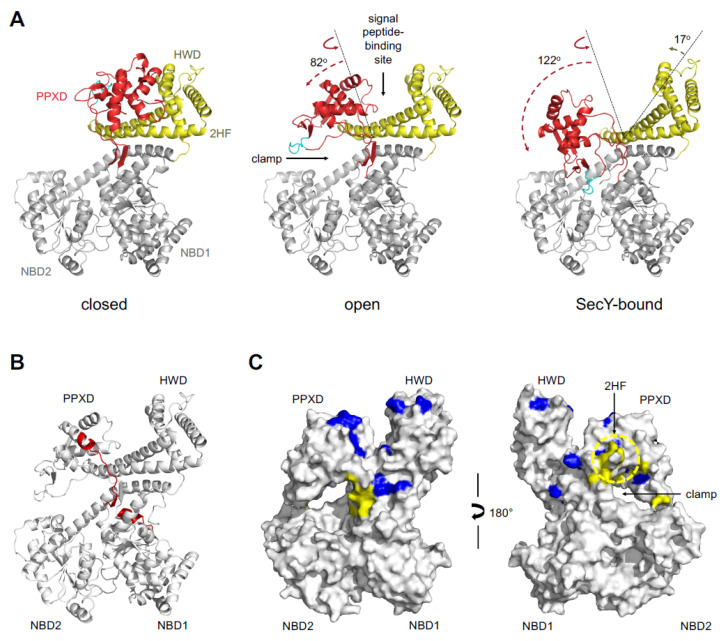
Conformational modeling of CDSecA2. (**A**) The closed conformation refers to our apo/ATP-γ-S-bound CDSecA2 structure, while the opened and SecY-bound conformation refer to the CDSecA2 models build using *B. subtilis* SecA (PDB ID 3JV2 [19]), and *T. maritima* SecA in complex with the SecY channel (PDB ID 3DIN [24]) as templates, respectively. The highly conserved loop of the PPXD contacting NBD1 and NBD2 in the SecY-bound CDSecA2 conformation complex is highlighted in cyan. The PPXD (red) and HWD-HSD (yellow) translations (dashed line arrows) and rotations (solid line arrows) compared to their positions in the closed structure (dotted line) are depicted. (**B**) Regions of the open CDSecA2 model lacking a template (P220-L236 and Q357-E364) for the SecY-bound CDSecA2 model are shown in red, and the remaining regions are shown as white ribbons. (**C**) The surface of the CDSecA2 model in an open conformation with surface residues previously shown to be involved in signal and nascent protein interaction in other SecA proteins is presented in two different orientations. The sequence alignment in Appendix A was used to depict the location of these residues with the corresponding color coding. Residues implicated in the signal peptide interaction are shown in blue [6,7,8,9,10], and residues implicated in the nascent protein interaction are shown in yellow [19,44,45]. The clamp that encloses the protein target is depicted with an arrow, and the 2HF loop is encircled with a yellow dotted line.

**Figure 4 ijms-21-06153-f004:**
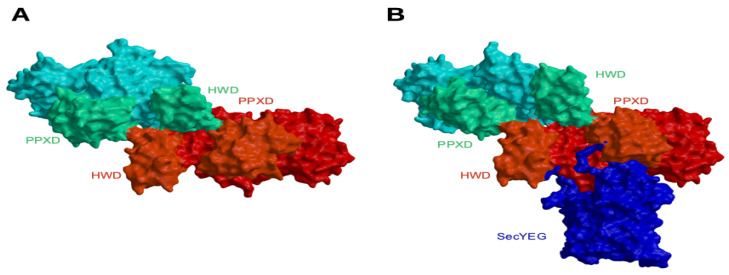
Model of the CDSecA2 dimer. (**A**) CDSecA2 dimer in the open conformation modeled based on *B. subtilis* SecA (PDB ID 3JV2) with one CDSecA2 molecule shown in red and the other in cyan. Interacting PPXDs and HWDs are highlighted. (**B**) CDSecA2 dimer in the SecY-bound conformation modeled based on *T. maritima* SecYEG (blue)-bound SecA dimer (PDB ID 3DIN). CDSecA2 dimer color coding is the same as in A.

**Figure 5 ijms-21-06153-f005:**
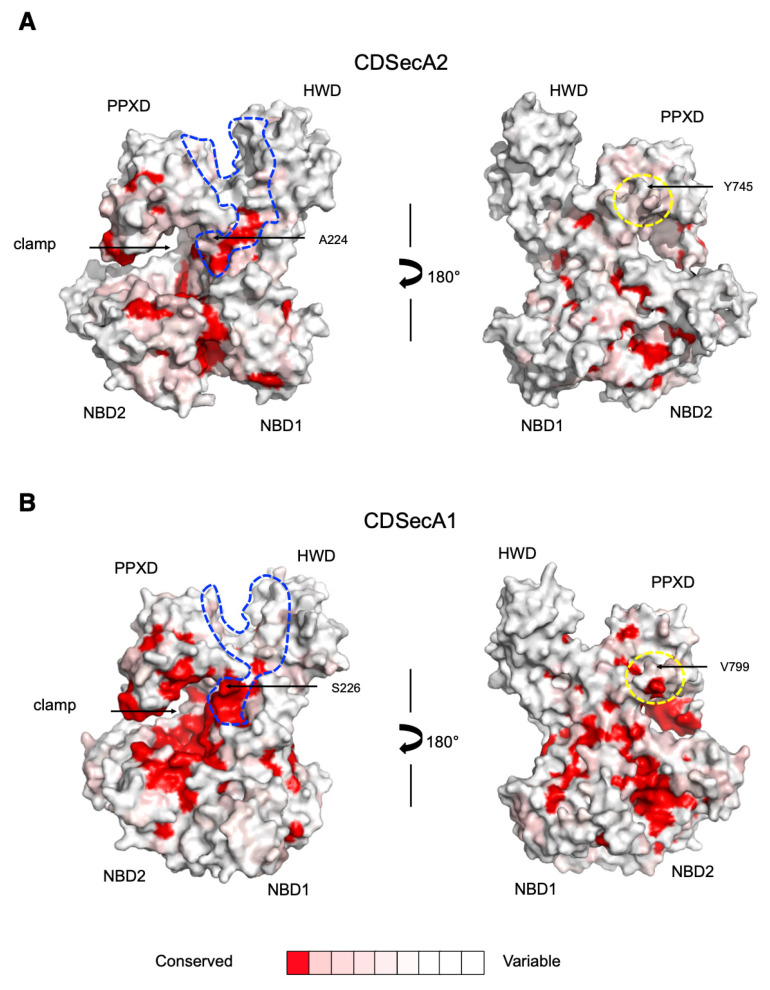
Surface conservation of SecA proteins in SecA2-encoding organisms. (**A**) Conservation of the SecA2 surface is shown based on the SecA2 structure of *C. difficile* and 30 of its orthologues. (**B**) Conservation of the SecA1 surface is shown based on the CDSecA1 model and 30 of its orthologues. Both models are shown in an open conformation and were built using *B. subtilis* SecA (PDB ID 3JV2) as a template. The clamp enclosing the protein target is depicted with an arrow, the signal-peptide binding groove is encircled with a blue line, and the 2HF loop is indicated with a yellow circle. Residues with the highest conservational scores are shown in red, and the color coding is shown at the bottom of the figure.

**Table 1 ijms-21-06153-t001:** Crystallographic statistics.

Crystallization and Data Collection		
Crystal	CDSecA2	CDSecA2 with ATP-γ-S
Unit cell		
a, b, c (Å) α, β, γ (deg)	a = 84.986, b = 96.847, c = 114.739 α = β = γ = 90.0°	a = 82.940, b = 96.837, c = 114.680 α = β = γ = 90.0
Space group	P2_1_ 2_1_ 2_1_ (number 19)	P2_1_ 2_1_ 2_1_ (number 19)
Molecules per au	1	1
Wavelength (Å)	0.9184	1.0000
Resolution range (Å)	48.4–2.3	50–2.6
No. of unique reflections	42821	29038
Completeness (last shell) (%)	99.8 (98.8)	99 (96.9)
Multiplicity	7.0	6.32
R meas (last shell) (%)	22.0 (163.3)	19.9 (171)
CC (1/2) (last shell) (%)	99.8 (61.9)	99.6 (80.8)
I/σ (last shell)	8.0 (1.2)	7.1 (1.33)
Refinement		
PDB ID	6SXH	6T4H
Resolution range (Å)	47.5–2.3	49.34–2.9
No. of reflections in working set	42717	21017
No. of reflections in test set	42717	21017
R-work	0.243	0.335
R-kick	0.272	0.380
RMSD from ideal geometry		
Bond length (Å)	0.019	0.014
Bond angles (°)	2.07	1.78
No. of atoms in au	8578	7720
Protein atoms	7621	7635
Water molecules	315	42
Mean *B* value (Å^2^)	29.32	48.49
Ramachandran plot statistics		
Favored	727 (94.8%)	644 (83.7%)
Allowed	38 (5.0%)	88 (11.4%)
Outliers	2 (0.3%)	37 (4.8%)

**Table 2 ijms-21-06153-t002:** Superposition of CDSecA2 and other SecA molecules. All structures were superimposed onto the CDSecA2 molecule with flexible structure alignment by chaining aligned fragment pairs allowing twists using software FATCAT [25]. PPXD and C-terminal tails were excluded from the alignment. * marks the structure shown in Figure 1B or Figure 1C.

PDB ID	Molecule-in Complex with	Species	Conformation	RMSD (Å) (Sequence Length/Identity (%))
6SXH *	SecA2	*C. difficile*	Closed	
6T4H	SecA2-ATP-γ-S	*C. difficile*	Closed	0.85 (768/100)
1NKT	SecA1-ADP-Mg	*M. tuberculosis*	Closed	1.78 (621/46)
1M6N *	SecA1	*B. subtilis*	Closed	2.31 (624/50)
4UAQ *	SecA2	*M. tuberculosis*	Intermediate	3.03 (563/32)
2IBM	SecA1-ADP	*B. subtilis*	Closed	3.07 (625/49)
3JV2 *	SecA1-peptide	*B. subtilis*	Open	2.10 (621/50)
6ITC	SecA1-SecYEG-peptide	*B. subtilis*	YEG-bound	2.30 (624/50)
5EUL	SecA1- SecY-peptide	*B. subtilis*	YEG-bound	2.51 (572, 45)
3DIN *	SecA1-SecYEG	*T. maritima*	YEG-bound	3.13 (597/43)

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
