# Peer review of "The Structure of *Clostridioides difficile* SecA2 ATPase Exposes Regions Responsible for Differential Target Recognition of the SecA1 and SecA2-Dependent Systems"

_ijms, 2020, doi:10.3390/ijms21176153_

Round 1

Reviewer 1 Report

The paper by Lindič and others reports a thorough study of the SecA2 ATPase from the Gram-positive bacterium Clostridioides difficile (CDSecA2). The reported crystal structures of the SecA2 apo and ATP-γ-S-bound form reveal in detail the closed conformation of the protein and comparison with SecA2 and SecA from other organisms, together with modelling and with evolutionary conservation analysis provide important new information about CDSecA2. All the experiments have been carefully planned and competently performed.  The results are sound and properly reported in the manuscript. For the above reasons I recommend the publication of this manuscript.

I have only a very minor point. This regards Figure 2B, reporting the preprotein cross-linking domains in different conformations. In my opinion, this figure will be better understood if largely different colors will be used to represent the conformational variability of such domains or, even better, if ribbons, like those shown in Figure 2C will be used instead of space-filling.

Some mistyping is present. E.g. MgCl2 instead of MgCl2 in paragraph 4.2 and in paragraph 4.3: MgAc2 instead of Mg(Ac)2, or, better, of the whole formula Mg(CH3COO)2.

Author Response

Reviewer 1

The paper by Lindič and others reports a thorough study of the SecA2 ATPase from the Gram-positive bacterium Clostridioides difficile (CDSecA2). The reported crystal structures of the SecA2 apo and ATP-γ-S-bound form reveal in detail the closed conformation of the protein and comparison with SecA2 and SecA from other organisms, together with modelling and with evolutionary conservation analysis provide important new information about CDSecA2. All the experiments have been carefully planned and competently performed.  The results are sound and properly reported in the manuscript. For the above reasons I recommend the publication of this manuscript.

I have only a very minor point. This regards Figure 2B, reporting the preprotein cross-linking domains in different conformations. In my opinion, this figure will be better understood if largely different colors will be used to represent the conformational variability of such domains or, even better, if ribbons, like those shown in Figure 2C will be used instead of space-filling.

Answer #1:

As the reviewer 1 suggested, we have modified the colors on the Figure 1B, depicting the PPXD domain conformations (see page 4). We decided to leave the surface representation after trying and realizing the ribbon representation was too crowded. We used yellow, green, violet and orange color for different PPXD conformations. The colors now fit those used for the same molecules presented in the panel C. We have also amended the figure legend describing the usage of colors accordingly (see page 5). The text in the figure legend now reads: “Closed, intermediate and open conformations of CDSecA2 (PDB ID 6SXH), M. tuberculosis SecA2 (PDB ID 4UAQ) [23] and of B. subtilis SecA (PDB ID 3JV2) [19], are shown in yellow, green and violet, respectively.”

Some mistyping is present. E.g. MgCl2 instead of MgCl2 in paragraph 4.2 and in paragraph 4.3: MgAc2 instead of Mg(Ac)2, or, better, of the whole formula Mg(CH3COO)2.

Answer #2:

We amended both formulas as suggested.

Reviewer 2 Report

In this manuscript, the authors present a structural biology exploration of the SecA2 protein from Clostridioides difficile. By obtaining crystal structures of both the protein’s apo and ATP-γ-S-complexed forms, they are able to identify areas of the protein involved in its target recognition and its role in interactions with the SecY channel. All structures and models are presented very well and with excellent consideration for biological interpretation (e.g., overlays in Fig.1C are very useful for understanding differences in SecA1/A2 across species). Such investigations will absolutely be of interest to those interested in bacterial secretory proteins and C. difficile in particular. With that in mind, the manuscript could benefit from a more detailed discussion of the findings’ relevance (e.g., how does a better understanding of SecA2 structure in C. difficile and how it compares to SecA1 support further work, and how may it impact the community’s ability to combat C. diff. as a pathogen). The manuscript text is of excellent quality – I enjoyed reading it, despite finding most structural biology papers rather dry!

 Major concerns:

 It is not entirely clear from the introduction what the knowledge gap is or exactly which features are of interest to the study. It certainly provides an excellent background description regarding SecA and SecA2 proteins. I don’t believe any further work is necessary but further context provided in the text would be helpful.

 Minor comments:

Thank you for making the PDB codes obvious!

The final sentence of the Discussion includes the phrase “This knowledge will enable to address”, which sounds a bit awkward. It could be rephrased as “This knowledge will enable us to address”.

Author Response

In this manuscript, the authors present a structural biology exploration of the SecA2 protein from Clostridioides difficile. By obtaining crystal structures of both the protein’s apo and ATP-γ-S-complexed forms, they are able to identify areas of the protein involved in its target recognition and its role in interactions with the SecY channel. All structures and models are presented very well and with excellent consideration for biological interpretation (e.g., overlays in Fig.1C are very useful for understanding differences in SecA1/A2 across species). Such investigations will absolutely be of interest to those interested in bacterial secretory proteins and C. difficile in particular. With that in mind, the manuscript could benefit from a more detailed discussion of the findings’ relevance (e.g., how does a better understanding of SecA2 structure in C. difficile and how it compares to SecA1 support further work, and how may it impact the community’s ability to combat C. diff. as a pathogen). The manuscript text is of excellent quality – I enjoyed reading it, despite finding most structural biology papers rather dry!

Answer #1:

To accommodate reviewers request, we have modified and extended the final paragraph of the discussion where we have described our findings relevance in more detail (see page 13). The paragraph now reads:

“To summarize, the analyses presented here identified regions that contribute to the mechanism of SecA2-specific target recognition and interaction with the SecY channel. Our modeling exposed importance of the open conformation for formation of dimer which can bind the SecY. An immediate suggestion from this study is that locking PPXD domain in closed or transient conformation by a compound binding in the hinge region connecting PPXD domain to NBD1 may provide a new target site for drug discovery. Moreover, we were unable to find a template to model this region in the open conformation (Figure 3). This suggests that a ligand binding in that area may specifically inhibit translocation of proteins in C. difficile SecA2 pathway, and due to the almost identical sequence in this regions, also its SecA1 homologue. Not so surprisingly, this further suggests that mechanistic differences between SecA1 and SecA2 should be studied pairwise. Nevertheless, our analysis (Figure 5) exposed lower sequence conservation in the CDSecA2 substrate loading region (encircled with blue line), and differences in 2HF finger which pinpoint to further mutation studies to be tested in a substrate and SecY binding assay. Hence, we expect that the knowledge gathered in this study will enable us to address the question of how the SecA1 protein functions a part of the “housekeeping” export system in bacteria and, furthermore, how the SecA2 protein functions as a part of the translocation system for specific, possibly folded protein substrates [2–4, 6–10, 36].“

Major concerns:

It is not entirely clear from the introduction what the knowledge gap is or exactly which features are of interest to the study. It certainly provides an excellent background description regarding SecA and SecA2 proteins. I don’t believe any further work is necessary but further context provided in the text would be helpful.

Answer #2:

To describe the knowledge gap and the structural features of interest for the CDSecA2, we have changed and expanded the last paragraph of the introduction (see top of page 3), that now reads:

“The general understanding of roles of SecA1 and SecA2 in protein translocation has been established, however, insight in the specific differences called for structural analysis. We therefore determined two crystal structures of the CDSecA2 protein (apo and in complex with adenosine-5'-(γ-thio)-triphosphate (ATP-γ-S)). We confirmed the ATPase activity of CDSecA2 whereas the analysis combining sequential and structural data exposed importance of open conformation for dimerization and SecY binding, and sites relevant for further mechanistic studies of differences in interactions of SecA2 and SecA1 with their substrate and SecY channel.“

Minor comments:

Thank you for making the PDB codes obvious!
The final sentence of the Discussion includes the phrase “This knowledge will enable to address”, which sounds a bit awkward. It could be rephrased as “This knowledge will enable us to address”.

Answer #3:

The sentence (see page 14) was amended accordingly and in respect with the expansion of the last paragraph in discussion, and now reads:

“ Hence, we expect that the knowledge gathered in this study will enable us to address …”

In addition, we have noticed and amended two small mistakes in the text:

- page 2, second paragraph: last sentence was divided into two sentences for clarification, and now reads:

“Interestingly, SecA1 and SecA2 are both essential for C. difficile survival [12]. An essential function of the SecA2 system was only observed in one other bacterium, Corynebacterium glutamicum [13].

- page 9, legend for Figure 2, misspelling in the last sentence was amended: “dotte dline” was changed to “dotted line”.